# HPV Opportunistic Vaccination: A Literature Review and a Single-Center Experience in Northern Italy through the COVID-19 Pandemic

**DOI:** 10.3390/vaccines11091435

**Published:** 2023-08-31

**Authors:** Francesco Cantatore, Nadia Agrillo, Alessandro Camussi, Massimo Origoni

**Affiliations:** Department of Gynecology & Obstetrics, Vita Salute San Raffaele University School of Medicine, IRCCS Ospedale San Raffaele, Via Olgettina 60, 20132 Milano, Italy; agrillo.nadia@hsr.it (N.A.); camussi.alessandro@hsr.it (A.C.)

**Keywords:** HPV, vaccine, opportunistic, cervical cancer, COVID-19 pandemic, 9vHPV vaccine

## Abstract

The World Health Organization (WHO) set the goal of 90% HPV vaccination coverage in the population to eliminate cervical cancer. Opportunistic vaccination is performed outside the free vaccination or catch-up programs. Both free and opportunistic HPV vaccination programs experienced slowdowns during the COVID-19 pandemic. In this retrospective study, we aimed to identify the benefits and the obstacles of opportunistic vaccination among male and female individuals who took advantage of the “on-demand” service offered by San Raffaele Hospital in Milan from April 2018 to May 2023. The impact that the COVID-19 pandemic had on vaccination adherence was also analyzed. Data on a total of 527 subjects were collected from an in-house database and through personal interviews. Women in the cohort of older patients (over 25) adhered to the vaccination schedule more than younger women. Opportunistic vaccination request is influenced by the need of a gynecologist, a general practitioner, or public health clinic availability. Women also showed good adherence to screening, demonstrating awareness of the importance of cervical cancer secondary prevention despite vaccination. Opportunistic vaccination offers the possibility of including individuals excluded from the free vaccination campaigns, often already affected by lesions caused by HPV, providing increased viral clearance and faster lesion regression. The main limit remains the economic burden.

## 1. Introduction

Human papillomavirus (HPV) infection, to date, is the most common sexually transmitted disease, and the leading cause of cervical cancer, and a fraction of other anogenital and oropharyngeal malignancies [1,2,3], and it is related to several non-malignant conditions such as genital warts and the juvenile occurrence of recurrent respiratory papillomatosis [4]. HPV vaccines have been developed against several HPV types: all of them have been shown to be safe, highly immunogenic, and effective against HPV infection and high-grade cervical lesions [5]. They are mostly effective when administered to HPV-naïve individuals. However, catch-up vaccination of higher age cohorts may be cost-effective, although the cost-effectiveness generally decreases with increasing age at vaccination [6].

Countries that achieved high vaccination coverage of the target population have observed a 73–85% decrease in the prevalence of HPV positivity for viral strains included in the vaccine and a 41–57% decrease of high-grade cervical lesions (HSIL) among young women [7]. A burden of 90% HPV vaccine coverage in the target population is the key to achieve the ambitious goal recently set by WHO [8] to eliminate cervical cancer [9]. However, HPV vaccine coverage varies by region and income level around the world [10]. Nowadays, it is known that vaccination, in addition to its role as primary prophylaxis, also acts with an adjuvant role, when administered following conization for high-risk lesions, significantly decreasing the rate of recurrences [11].

In Italy, the HPV female vaccination campaign began in all regions by the end of 2008 with the goal of achieving 95% vaccination coverage within the following 5 years. The Emilia Romagna region collected data on the progress of the HPV vaccination program by region and birth cohort as of 31 December 2008 and 30 June 2009. Data analysis showed marked heterogeneity for vaccine coverage across Italian regions. Mean rates of adherence with at least one dose of vaccine were 61.8% and 43.9% for the 1997 and the 1996 cohorts, respectively. As for vaccination coverage with three doses, mean rates were 34.5% and 26.7% for the 1997 and the 1996 cohorts, respectively [12]. Starting in 2015, the vaccination campaign was also extended to males in several Italian regions. This allowed demonstrating how the response against the infection occurs in males as much as in females, managing to reduce the prevalence of HPV type 16 in 36% of males versus 21% of females. This result is important in the context of minimizing the reservoir of infection, represented by males [12].

The concept of opportunistic vaccination is part of the increased knowledge and awareness that has been achieved in recent years; it is performed outside the free vaccination campaign or catch-up programs. In general, individuals asking to receive opportunistic vaccination are offered the three-dose scheduled program at a so-called social price.

Different approaches for population and individual HPV vaccination programs have been chosen according to the specificities of national/regional healthcare systems. Sociodemographic factors such as educational level, socioeconomic status, personal beliefs, and immigration status may be associated with attitudes towards primary prevention and vaccines, and thus towards HPV vaccine uptake [13,14]. From this viewpoint, the knowledge of the social profile and attitudes of individuals referred to HPV vaccine uptake on a voluntary basis is particularly interesting to assess the extent to which social equity is achieved, and to better manage the healthcare decision-making process.

Two Scandinavian population studies investigated the sociodemographic characteristics of the population and the rate of adherence to opportunistic vaccination. The overall initiation rate of opportunistic HPV vaccination consistently increased with year of birth, parental education level, and family income; on the other hand, having two immigrant parents or a parental low-prestige occupational level was strongly associated with lower opportunistic HPV vaccination adherence. Free HPV vaccination offered in schools achieved the highest uptake and showed the lowest disparity according to country of birth and socioeconomic background of parents [15,16].

Another survey has been recently performed in Italy by the Veneto region, reporting the results of a survey conducted among young women from age-at-birth cohorts not included in active vaccination campaigns. Compared with unvaccinated women, higher adherence rates to invitation for cervical cancer screening and a lower CIN3+ diagnosis was observed among vaccinated women. Age at vaccination was inversely related to vaccination efficacy [17].

This trend appears to be in line with what was shown in two Swedish studies whose results showed that vaccinated women responded to screening more than unvaccinated women (74.4% vs. 69.9%, *p* < 0.001), and that women vaccinated against HPV were more likely to have a university education, be in the highest income quartile and be native to the nation [18]. Screening adherence after three years of follow-up was higher in the vaccinated against HPV. Mostly, the differences in participation were explained by socioeconomic differences between vaccinated and unvaccinated women [19]. In addition, compared to unvaccinated women, HPV opportunistically vaccinated women were just as likely, if not more likely, to participate in the cervical screening program following invitation [20]. The increased prevalence of Pap negative tests after the completion of the vaccination cycle, 62% of the total vaccinated with three doses, supports the theory that the vaccine would help in the negativization of HPV lesions [20]. A recent Italian article showed a high statistically significant difference (*p* < 0.0001) of the cytological negativization time after exposure to the third dose of vaccine in women who started HPV vaccination with an abnormal Pap test and also in women who started HPV vaccination with an hrHPV+ test, concluding that 9vHPV vaccination can play a positive role in shortening the clearance time of hrHPV+ or Pap smear positivity in sexually active adult women [21].

HPV vaccination programs have experienced significant slowdowns during the COVID-19 pandemic of about 15% and, in particular, a study conducted in the United States that examined the impact of the first wave of the COVID-19 pandemic reported that HPV vaccine doses administered declined by a median of 63.6% among children 9–12 years and 71.3% among teenagers, compared with the 2 years preceding the pandemic [22]. This significantly reduced the opportunity for many adolescents to be vaccinated in school or other public settings. The consequences of missed vaccinations include an excess of preventable cases of genital warts, cervical cancer, and other HPV-related diseases under detection. A study conducted in Switzerland and Greece estimates that, in order to eliminate the dose deficit, monthly vaccination rates should be increased by 6.3% and 6.0%, respectively [23]. Although a slow recovery in vaccination rates has been observed globally, the vaccination gap accumulated during the COVID-19 pandemic persists in many European countries.

The reported literature data are summarized in Table 1.

## 2. Materials and Methods

We conducted a retrospective study including the whole number of male and female individuals who took advantage of the “on-demand” opportunistic vaccination service offered by San Raffaele Hospital in Milan from April 2018 to May 2023, with the aim of identifying the benefits associated with such an offer and what the obstacles have been to achieving the goals set by the WHO Global Strategy to Accelerate Cervical Cancer Elimination [8].

Both men and women of all ages who received 1, 2 or all 3 scheduled administrations of Gardasil9^®^ (MSD Vaccines) vaccine were included in the survey. Biographical data were extracted from the in-house database of the HPV vaccination service and a personal interview was conducted, asking participants the motivation and reason for which they required HPV opportunistic vaccination, adherence or non-adherence to cervical screening before and after vaccination, whether they ever had cervical interventions, the reason for previous non-adherence to the nationwide organized vaccination campaign in adolescence, and the reason for dropout for those who did not complete the vaccination cycle on May 2023. Men were asked why they decided to be vaccinated and whether they ever had an HPV-related injury or disease, and why they eventually dropped out from the organized vaccination program if they did not complete the vaccination cycle. Written consent was administered and signed before the first dose of vaccine by each subject. We also analyzed the impact that COVID-19 pandemic had on vaccination adherence and completion.

Descriptive statistical analysis of population data was performed using R software (available at: https://www.rproject.org) (accessed on 30 June 2023), version 4.2.0. Statistical analysis of results data was performed with Statistical Calculator software (available at: https://www.socscistatistics.com) (accessed on 30 June 2023) using the chi-square contingency table test, assuming a *p* value < 0.05 as significant.

## 3. Results

A total of 527 subjects, including 467 women (89%) and 60 men (11%) adhered to the opportunistic HPV vaccination offered by San Raffaele Hospital in Milan since April 2018, all of whom were residents of Lombardy region. Out of the whole group, 92 (17.4%) received one dose of the vaccine, 103 received two doses (19.5%), and 332 (62.9%) completed the vaccine cycle. We divided users by age, to understand which segment of the population was most interested and willing to receive an opportunistic vaccination offer. Of the total group, 59 subjects (11.1%) were under 25 years, 371 (70.3%) were between 25 and 45 years, and 97 (18.4%) were over 45 years. Out of the 527 subjects, 332 (62.9%) completed the three-dose vaccination cycle: 294 women (88.5%) and 38 men (11.4%). Also in this subgroup, the breakdown by age group mirrors what was obtained in the whole setting; in detail, 35 (10.5%) under 25, 257 (77.4%) between 25 and 45, and 40 (12%) over 45 completed the vaccination schedule.

### 3.1. Retrospective Survey of the Female Sample

Out of the 294 women who completed the vaccination offer, 22 (6.62%) belonged to the “under 25” category and 15 of them (68.1%) participated in an interview focused on the motivations for being opportunistically vaccinated: 9 requested the HPV vaccine because of the vaccine advertising media campaign, 3 because of a gynecologist’s recommendation, and 3 because of parental choice. Only two of them had already had a cytological cervical screening performed before; seven did not receive the vaccine at age 12 based on pediatrician’s advice, and eight because of parental choice. In this same group, 272 (81.9%) belonged to the “over 25” category and 186 (68.3%) of them participated in the interview: 18 requested the vaccine because of the vaccine advertising media campaign, 4 because were physicians, and 164 followed the recommendation of a gynecologist. According to their cervical screening attitude, 5 (2.6%) had never had a previous Pap test, 46 (24.7%) cytologically tested negative before vaccination, and 135 (72.5%) reported a non-negative Pap test; finally, 55 (40.7%) had previously undergone conservative cervical treatments (LEEP/ablative cervical treatment). After the completion of the three-dose vaccination schedule, among the 135 women with a positive Pap test, 13 (9.6%) were still not re-tested, 84 (62.2%) re-tested negative, and 38 (28.1%) re-tested positive again. Among the 173 (88.7%) women who did not complete the three-dose schedule, 47 (27.1%) are still undergoing the vaccine cycle, 30 (17.3%) have completed the vaccine cycle at other vaccination centers, 74 (42.7%) have discontinued the vaccination cycle due to the COVID-19 pandemic, and 22 (12.7%) have discontinued the cycle for personal reasons. In this group, 16 (9.2%) women belonged to the “under 25” category, and 10 underwent vaccination due to parental suggestion, 3 because of the vaccine advertising campaign, 2 due to pediatrician/general practitioner recommendation, and 1 because of the recommendation of a gynecologist. None of them already had a Pap test before vaccination. Seven subjects of the “under 25” group were not vaccinated by the age of 12 because of pediatrician advice and nine due to parental choice. In this same group of uncomplete schedule, 157 (90.7%) women belonged to the “over 25” category: 7 were vaccinated due to the vaccine advertising campaign, 132 on the advice of a gynecologist, and 18 following media communication. Among the “over 25” group, 7 (4.4%) never had a Pap test before vaccination, 59 (37.5%) had a negative Pap test, and 91 (57.9%) had a positive Pap test. Finally, 26 (28.5%) had already undergone cervical surgery (LEEP/ablative cervical treatment). Out of the 91 women with a positive Pap test before vaccination, 71 (78%) were still not re-tested, 16 (17.5%) had a negative Pap test, and 4 (4.3%) still had a positive Pap test.

### 3.2. Retrospective Survey of the Male Sample

Thirty-eight men completed the vaccination cycle: 13 (3.91%) in the “under 25” category and 25 (7.53%) in the “over 25” category. Among men under 25 years of age, four performed the vaccine based on a pediatrician’s instructions and nine following parental choice. None of them had HPV lesions at the time of the opportunistic vaccination schedule. In the “over 25” subgroup, 19 participated in the interview, and out of them 6 received the vaccine based on the dermatologist’s recommendation, 9 due to sexual partner genital infectious conditions, and 4 on a urologist’s recommendation; 6 had no HPV lesions and 13 were diagnosed HPV lesions. Of the 22 men who did not complete the three-dose schedule, 3 are still on the vaccination cycle, 12 discontinued the cycle due to the COVID-19 pandemic, 4 completed the vaccine cycle at other vaccine centers, and 3 discontinued the cycle for personal reasons. In this subgroup, 8 (36.3%) belonged to the “under 25” category and all of them had performed the vaccination by maternal choice. None had previous HPV lesions. Fourteen (63.6%) men belonged to the “over 25” category, and out of them five performed the vaccination following a dermatologist’s recommendation, seven for a sexual partner genital infection, and two on a urologist’s recommendation. Eight of them had HPV lesions; six had no HPV lesions. 

### 3.3. COVID-19 Pandemic Effect

The trend of population turnout for opportunistic HPV vaccination underwent major changes with the onset of the COVID-19 pandemic; we observed a sharp decline in vaccination in 2020 and many unfinished vaccination cycles. In our single-center case series, 101 vaccine doses were administered in April–December 2018, 162 vaccine doses in 2019, 88 in 2020, 164 in 2021, 167 in 2022, and 95 in January–May 2023. At the very beginning of the COVID-19 pandemic, and in particular during the generalized lock-down period, which in Italy went from March to December 2020, a 45.68% decrease in opportunistic HPV vaccine administrations was recorded at our institution. The whole set of results from our study is reported in Table 2.

## 4. Discussion

This retrospective analysis examined some of the aspects underlying the choice for adherence to the HPV opportunistic vaccination offer. We noted that women in the cohort of older patients (over 25) adhered to and completed the vaccination schedule more than younger women probably due to increased awareness of the health issue and the possibility of bearing the cost of vaccination. Analysis of the data we collected shows that opportunistic vaccination appears to be influenced by the need of a gynecologist, general practitioner, pharmacy or public health clinic availability and attendance, which in part explains why the completion rate is in general lower than that obtained in a school setting where vaccination is more easily accessible. None of the women interviewed, either under 25 or over 25, received directions from their general practitioner, but only from gynecologists or through social media campaigns.

In this regard, several research surveys have been conducted in different countries worldwide, assessing women’s awareness of vaccination and the contribution made by healthcare providers. An Australian study in 2012 investigated the counseling of general practitioners through a mixed method consisting of a questionnaire of knowledge and attitudes, and an audit assessing the overall vaccination rates since the beginning of the program; results showed that basic physicians have a good level of knowledge of HPV vaccination and are strongly committed to suggesting it for women up to 27 years, but are much less so for women aged 27 to 45 years; from 2007 to 2010, only 1.9% of women aged 27–45, who attended basic physician appointments, began an opportunistic vaccination program against HPV. Of these, however, the majority completed the three-dose schedule. Female general practitioners had significantly higher vaccination rates among their patients [25]. Another report from Australia determined the percentage of women attending family planning clinics in New South Wales (FPNSW) aged ≤26, who were aware of the free HPV vaccination program and had received a full course of the vaccine or least one injection. Data showed that 83% of women had knowledge about the HPV vaccine and 56% attended a basic physician for at least one visit; most women (72.4%) visited a basic physician in the previous 6 months. Overall, basic physicians suggested the vaccine to 110 (37.4%) women during a recent visit; 59 (53.6%) of these women attended their general practitioner specifically to undergo HPV vaccination. Of the 179 who answered the question about awareness of the availability of a free cycle of HPV vaccine, 76 (42.5%) were unaware that they could get free vaccination through their general practitioner [26]. It is quite clear that general practitioners should therefore use opportunistic visits by their young women patients to provide information on the HPV vaccination recovery programs and encourage them to participate in the opportunistic vaccination.

As far as it regards which interventions must be implemented to increase women’s adherence to cervical cancer prevention, an update of the 2011 Cochrane review was published. Overall, there was moderate-certainty evidence to suggest that personalized invitations are more likely to be successful. Low-certainty evidence supported the use of educational materials (RR 1.35, 95% CI 1.18 to 1.54; 63,415 participants; 13 studies) and lay health worker involvement (RR 2.30, 95% CI 1.44 to 3.65; 4330 participants; 11 studies) [27].

Specialists and family doctors should therefore be the first allies in the field of primary prevention and the first promoters of vaccination campaigns, both organized and opportunistic.

The sample of women we analyzed showed good adherence to screening, both before and after vaccination, demonstrating good awareness of the importance of continuing prevention checks despite vaccination. This is in line with other northern Europe population studies, underlining that raising awareness of health issues associated with HPV infection allows for the implementation of primary and secondary prevention strategies [15,16,17,18].

Our data show that opportunistic vaccination is a desirable tool for women diagnosed with cervical HPV lesions, in view of its demonstrated efficacy in normalizing cytologic changes and accelerating viral clearance [11], and reflect the recent guidance from the Italian National Institutes of Health regarding the recommendation to vaccinate all women treated for severe cervical dysplasia (CIN2+) [12].

Male participation is in general still significantly lower than female participation, despite robust evidence of significant advantages and benefits of vaccinating men, as reported in a review article in 2021 [28]. However, after the introduction of national vaccination programs against HPV for women and girls in most developed countries, as it regards male vaccination, few countries have established national programs; several authors underlined the crucial importance of monitoring the impact of the HPV vaccine in men and the benefits that occur, informing and disseminating data and figures to implement such vaccination programs worldwide [29]. Our case records show that opportunistic vaccination is a valuable tool available for the adult males or young men who for some reasons did not benefit from the organized HPV vaccination campaign at school age. Analysis of our data reveals that most men accessing opportunistic vaccination have been diagnosed with HPV lesions either themselves or in their sexual partners.

The ESGO/EFC position paper of the European Society of Gynaecologic Oncology and the European Federation for Colposcopy reiterates that gender-neutral vaccination provides direct protection for all men and improves vaccination coverage. School vaccination programs appear to be the most effective. An extensive opportunistic recovery or vaccination program for young adult women and men significantly improves effectiveness [30].

The COVID-19 pandemic caused significant disruptions to healthcare, including a reduction in the routinely recommended HPV vaccines in several European countries. The percentage of incomplete cycles was demonstrated also to be significant prior to the pandemic but the main cause of failure to complete the vaccination cycle, for both men and women, was the beginning of the COVID-19 pandemic and the consequent restrictions applied. This situation is in line with the global experience. Recent WHO/UNICEF estimates of vaccine coverage (WUENIC) showed a sharp decline in global vaccine coverage in 2021, with 25 million children losing life-saving vaccines. These dangerous decreases in coverage have been the largest sustained decline in infant vaccinations in about 30 years. As far as it regarded HPV vaccination, more than a quarter of the HPV vaccine coverage achieved in 2019 has been lost [31]. According to UNICEF data globally, only one in eight girls is vaccinated against human papillomavirus. Although the momentum was developing before the global pandemic, vaccination programs against HPV were severely affected by COVID-19. HPV vaccines are often delivered to schools because the primary target group for this vaccine is girls aged 9 to 14. Interruptions due to COVID-19 resulted in the closure of schools and health facilities and delayed vaccination cycles. In 2020 there were over 600,000 new cases of cervical cancer and 340,000 deaths. Ninety per cent of these cases and deaths occurred in low- and middle-income countries where access to prevention, screening, and treatment services is more limited, and almost 60% of these cases occur in countries that have not yet introduced vaccination against HPV. This finding further reinforces the importance of HPV vaccination in primary prevention. The good news is that 20 of the countries with the highest burden will introduce HPV vaccine within the next two to three years [31].

The analysis of data we have collected showed that the highest decrease in opportunistic HPV vaccination was recorded in our center during the COVID-19 pandemic lock-down period (March to December 2020). This is in line with the world literature as well as easily explained by the need to reduce deferrable health activities with the aim of limiting the contagion from COVID-19.

A survey conducted by the Italian Ministry of Health in mid-2020 showed a reduction in vaccination activities due to diversion of resources for the response to the COVID-19 emergency. In 68% of cases, vaccinations of adolescents were the most delayed due to the pandemic. Data from the 2020 vaccination coverage of children and adolescents confirm this impact. A cross-sectional retrospective study was performed on the general population and HIV subjects. Data on anti-HPV vaccine coverage in 2019, before the COVID-19 pandemic, and data from 2020, after the announcement of the pandemic status and lockdown, and the implementation of restrictive measures to contain the infection, were compared. The sample consisted of men and women older than 18 and PLWH (people living with HIV) subjects from the Sicily region. In 2019, 330 doses of HPV vaccine were administered to the general population and 66 doses in PLWH; in 2020 the figure decreased to 190 doses in the general population and 42 in PLWH subjects, with a decrease of 42% in the general population and 36% in PLWH compared to 2019 [24]. Regarding the published evidence related to available cervical cancer screening modalities for HIV-seropositive women in developing countries, a review to assess, synthesize, and document this issue reported that screening methods used for HIV-seropositive women are the same as for HIV-negative women, with varying clinical performance and diagnostic accuracy; the target will be to integrate cervical cancer screening programs into already existing HIV services to enable early detection and treatment in these patients [32].

## 5. Conclusions

Opportunistic vaccination offers the pivotal possibility to include a section of the population excluded from the free vaccination campaigns, often already affected by lesions caused by the papilloma virus, guaranteeing these subjects a significant chance of increased viral clearance and faster lesion regression. Opportunistic vaccination also allows recovering all male subjects for whom free vaccination was introduced years later than for young women. Moreover, it guarantees the possibility of a personalized management of the vaccination cycle, offering the patient the freedom to conclude the vaccination cycle at different facilities. The need for adequate counseling by healthcare providers remains a fundamental element for adequate information, knowledge, comprehension, acceptance, and adherence to the opportunistic vaccination offer. However, there is still a strong need for general practitioners and specialty professionals to be properly informed and involved, as they represent the first reference for patients in terms of health education, making a significant contribution to the dissemination of information.

The main limit, however, remains the economic burden, which represents the main discriminating element and possible starting point of subsequent sociodemographic analysis of users who benefit from HPV vaccination, as already conducted in other states. The major challenge remains to recover from the slowdown that occurred worldwide during the COVID-19 pandemic, eliminate the deficit, and bring up the trend achieved in the pre-COVID era, determining long-term positive effects on public health.

## Figures and Tables

**Table 1 vaccines-11-01435-t001:** HPV opportunistic vaccination in different countries.

% HPV Vaccination Decrease (COVID-19)		% Initiation Rate		% Completion Rate		% Screening Attendance		% Vaccination	
		Opportunistic	Catch-Up	Opportunistic	Catch-Up	Unvaccinated	Vaccinated	Unvaccinated	Vaccinated
**Italy** [17,24]	42					50.1	50.5		
PLWH [24]	36								
**Greece** [19]	21.7								
**Switzerland** [23]	24.4								
**Norway** [15]		2.2	46.2	72.1	73				
**Parental country of birth**									
two Norwegian parents		2.9	53.9	72.9	73.5				
two immigrant parents		0.6	21.1	65.2	68.3				
one foreign/one Norwegian		3.8	45.1	71	71.6				
**Maternal education**									
None/primary/low secondary		1.6	40.4	70.8	71.6				
Upper secondary/non-tertiary		2.3	52	74.2	74.1				
Undergraduate		4.5	58	73.1	73.2				
Postgraduate		8.3	57.9	68.9	70.6				
**Sweden** [16]		37.3	47.8	89.4	66	69.6	74.4		
**Education** [19]									
*Less than high school*								9.5	2.2
*High school*								40.2	36.1
*University*								41.6	60
**Income** [19]									
*Quartile 1*								25.1	13.4
*Quartile 2*								25	21.3
*Quartile 3*								25	29.6
*Quartile 4*								24.9	35.8
**Australia** [25,26]		11		91		63	79		
**USA** [22]									
*9–12 years*	63.6								
*teenagers*	71.3								

**Table 2 vaccines-11-01435-t002:** HPV opportunistic vaccination at San Raffaele Hospital from April 2018 to May 2023.

**No. of subjects**			527		
**Mean age (range)**			36 (15–66)		
**Vaccination decrease °**			45.68%		
	**Males**		**Females**		*p* < 0.05
	60 (11%)		467 (89%)		
	**<25 year**	**>25 year**	**<25 year**	**>25 year**	*p* < 0.05
	21 (3.98%)	39 (7.4%)	38 (7.22%)	429 (81.4%)	
**Males + females**	**<25 year**		**>25 year**		*p* < 0.05
	59 (11.1%)		468 (88.2%)		
	**Completion rate**				
	332 (62.9%)				*p* < 0.05
	13 (3.91%)	25 (7.53%)	22 (6.62%)	272 (81.9%)	
	**Interview participation (schedule completed)**				
	13 (3.91%)	19 (76%)	15 (68.1%)	186 (68.3%)	*p* < 0.05
	**Motivations for opportunistic vaccination**				
**Media campaign**	-	-	9 (60%)	18 (9.67%)	*p* < 0.05
**Specialist advice**	-	10 (52.6%)	3 (20%)	164 (88.1%)	
**Parental choice**	9 (69.2%)	-	3 (20%)	-	
**Being doctors**	-	-	-	4 (2.15%)	
**Pediatrician advice**	4 (30.7%)	-	-	-	
**Partner infection**	-	9 (47.3%)	-	-	
	**HPV lesions before vaccination**				
**Total**	249 (47.2%)				
**HPV lesions**	-	21 (51.2% *)	-	-	
**Abnormal Pap test**	-	-	2 (13.3% **)	226 (65.8% ***)	
**LEEP/ablation**	-	-	-	81 (35.8% ***)	

° March–December 2020; * % of male subjects with lesions who participated in the interview and received at least one vaccine dose (19 received 3 doses, 22 received 1–2 doses); ** % of female subjects with lesions under 25 who participated in the interview and received at least one vaccine dose (15 received 3 doses); *** % of female subjects with lesions over 25 who participated in the interview and received at least one vaccine dose (186 received 3 doses, 157 received 1–2 doses).

## Data Availability

Data may be available upon request to the corresponding authors.

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
