# Peer review of "HPV Opportunistic Vaccination: A Literature Review and a Single-Center Experience in Northern Italy through the COVID-19 Pandemic"

_vaccines, 2023, doi:10.3390/vaccines11091435_

Round 1

Reviewer 1 Report

The authors present a single-centre study of HPV opportunistic vaccination through the COVID -19 pandemic, but also extensively discussed other studies on the HPV opportunistic vaccination. The study itself is quite informative and provides a range of relevant information collected through questionnaires, as well as a link to clinical data on HPV screening and medical interventions.

Comments and recommendations:

1.     The authors noted that HPV vaccination programs declined during the COVID -19 pandemic, which is true for many vaccination programs worldwide. However, the data show that the number of vaccine doses was lower only in 2020, whereas the numbers in 2021 and 2022 were similar to or even slightly higher than in 2019. COVID -19 pandemic period needs to be clarified in the paper. If the 45.68% decline in opportunistic HPV vaccination refers only to 2020, this should be presented differently and not as a COVID -19 pandemic that lasted much longer.

2.     The paper is a mixture of research paper and literature review, which is sometimes confusing. While the first section follows the format of a research paper, the discussion only briefly addresses the major findings of the retrospective study conducted by the authors. Much of the section is devoted to detailed descriptions of other studies of opportunistic HPV vaccination and the COVID19 effect of vaccination programs–mostly without direct comparison with the original data obtained in the study.

I suggest that a detailed literature review be included in the Introduction section, where some of the studies described in the Discussion are mentioned already. The discussion could instead focus more on the actual results and their comparison with other recent studies in the field.

The English is generally good, but there are a few places where the meaning is not entirely clear.

Please restructure/modify the following sentences:

·       Page 2: This allowed to demonstrate how the response against the infection occurs in males as much as in females, managing to reduce the prevalence of HPV16 in 36% of males versus 21% of females.

·       Page 2: Sociodemographic factors such as educational level, socioeconomic status, personal beliefs, and immigration status may be associated with attitudes towards health primary prevention and vaccines, and thus towards HPV vaccine uptake [13,14].

·       Page 2: In Italy, the HPV vaccination campaign began in all regions by the end of 2008 with the goal of achieving 95% vaccination coverage within the following 5 years. Is this in females? Please clarify.

·       Please write out PLWH in full (page 8).

·       Page 9, line 362: Please replace women with a significant option of increased viral clearance” to a more appropriate term.

Author Response

Authors wish to thank reviewer 1 for the precious comments and their reply follows:

  1. clarification of the COVID-19 period in which the major decline in vaccination rates was observed has been made in the text
  2. Introduction and Discussion section have been emended according to reviewer's comment
  3. English language editing has been modified according to reviewer's suggestions

Reviewer 2 Report

While you have selected a critical issue, you can improve this Systematic review significantly by reporting other such reviews published and whether your findings corroborate with reported data. Here is a list that I am proposing for you to consider; not all may be relevant:

1: Reifferscheid L, Kiely MS, Lin MSN, Libon J, Kennedy M, MacDonald SE.

Effectiveness of hospital-based strategies for improving childhood immunization

coverage: A systematic review. Vaccine. 2023 Jul 25:S0264-410X(23)00857-5. doi:

10.1016/j.vaccine.2023.07.036. Epub ahead of print. PMID: 37500415.

2: Fragoulis GE, Dey M, Zhao S, Schoones J, Courvoisier D, Galloway J, Hyrich

KL, Nikiphorou E. Systematic literature review informing the 2022 EULAR

recommendations for screening and prophylaxis of chronic and opportunistic

infections in adults with autoimmune inflammatory rheumatic diseases. RMD Open.

2022 Nov;8(2):e002726. doi: 10.1136/rmdopen-2022-002726. PMID: 36323488; PMCID:

PMC9639159.

3: Meinderts JR, Prins JR, Berger SP, De Jong MFC. Follow-Up of Offspring Born

to Parents With a Solid Organ Transplantation: A Systematic Review. Transpl Int.

2022 Aug 5;35:10565. doi: 10.3389/ti.2022.10565. PMID: 35992748; PMCID:

PMC9389717.

4: Staley H, Shiraz A, Shreeve N, Bryant A, Martin-Hirsch PP, Gajjar K.

Interventions targeted at women to encourage the uptake of cervical screening.

Cochrane Database Syst Rev. 2021 Sep 6;9(9):CD002834. doi:

10.1002/14651858.CD002834.pub3. PMID: 34694000; PMCID: PMC8543674.

5: Navarro-Torné A, Curcio D, Moïsi JC, Jodar L. Burden of invasive group B

Streptococcus disease in non-pregnant adults: A systematic review and meta-

analysis. PLoS One. 2021 Sep 30;16(9):e0258030. doi:

10.1371/journal.pone.0258030. PMID: 34591924; PMCID: PMC8483371.

6: Jones JL, Tse F, Carroll MW, deBruyn JC, McNeil SA, Pham-Huy A, Seow CH,

Barrett LL, Bessissow T, Carman N, Melmed GY, Vanderkooi OG, Marshall JK,

Benchimol EI. Canadian Association of Gastroenterology Clinical Practice

Guideline for Immunizations in Patients With Inflammatory Bowel Disease

(IBD)-Part 2: Inactivated Vaccines. Gastroenterology. 2021 Aug;161(2):681-700.

doi: 10.1053/j.gastro.2021.04.034. PMID: 34334167.

7: Oku K, Hamijoyo L, Kasitanon N, Li MT, Navarra S, Morand E, Tanaka Y, Mok CC.

Prevention of infective complications in systemic lupus erythematosus: A

systematic literature review for the APLAR consensus statements. Int J Rheum

Dis. 2021 Jul;24(7):880-895. doi: 10.1111/1756-185X.14125. Epub 2021 May 17.

PMID: 33999518.

8: Benchimol EI, Tse F, Carroll MW, deBruyn JC, McNeil SA, Pham-Huy A, Seow CH,

Barrett LL, Bessissow T, Carman N, Melmed GY, Vanderkooi OG, Marshall JK, Jones

JL. Canadian Association of Gastroenterology Clinical Practice Guideline for

Immunizations in Patients With Inflammatory Bowel Disease (IBD)-Part 1: Live

Vaccines. Gastroenterology. 2021 Aug;161(2):669-680.e0. doi:

10.1053/j.gastro.2020.12.079. Epub 2021 Feb 19. PMID: 33617891.

9: Hutchinson AF, Smith SM. Effectiveness of strategies to increase uptake of

pertussis vaccination by new parents and family caregivers: A systematic review.

Midwifery. 2020 Aug;87:102734. doi: 10.1016/j.midw.2020.102734. Epub 2020 May

11. PMID: 32470666.

10: Vardanjani HM, Borna H, Ahmadi A. Effectiveness of pneumococcal conjugate

vaccination against invasive pneumococcal disease among children with and those

without HIV infection: a systematic review and meta-analysis. BMC Infect Dis.

2019 Aug 5;19(1):685. doi: 10.1186/s12879-019-4325-4. PMID: 31382917; PMCID:

PMC6683423.

11: Mapanga W, Girdler-Brown B, Feresu SA, Chipato T, Singh E. Prevention of

cervical cancer in HIV-seropositive women from developing countries through

cervical cancer screening: a systematic review. Syst Rev. 2018 Nov 17;7(1):198.

doi: 10.1186/s13643-018-0874-7. PMID: 30447695; PMCID: PMC6240280.

12: Tengan FM, Abdala E, Nascimento M, Bernardo WM, Barone AA. Prevalence of

hepatitis B in people living with HIV/AIDS in Latin America and the Caribbean: a

systematic review and meta-analysis. BMC Infect Dis. 2017 Aug 24;17(1):587. doi:

10.1186/s12879-017-2695-z. PMID: 28836955; PMCID: PMC5571507.

13: Teo E, Lockhart K, Purchuri SN, Pushparajah J, Cripps AW, van Driel ML.

Haemophilus influenzae oral vaccination for preventing acute exacerbations of

chronic bronchitis and chronic obstructive pulmonary disease. Cochrane Database

Syst Rev. 2017 Jun 19;6(6):CD010010. doi: 10.1002/14651858.CD010010.pub3. PMID:

28626902; PMCID: PMC6481520.

14: Dodd PJ, Prendergast AJ, Beecroft C, Kampmann B, Seddon JA. The impact of

HIV and antiretroviral therapy on TB risk in children: a systematic review and

meta-analysis. Thorax. 2017 Jun;72(6):559-575. doi:

10.1136/thoraxjnl-2016-209421. Epub 2017 Jan 23. PMID: 28115682; PMCID:

PMC5520282.

15: Ford N, Shubber Z, Meintjes G, Grinsztejn B, Eholie S, Mills EJ, Davies MA,

Vitoria M, Penazzato M, Nsanzimana S, Frigati L, O'Brien D, Ellman T, Ajose O,

Calmy A, Doherty M. Causes of hospital admission among people living with HIV

worldwide: a systematic review and meta-analysis. Lancet HIV. 2015

Oct;2(10):e438-44. doi: 10.1016/S2352-3018(15)00137-X. Epub 2015 Aug 11. PMID:

26423651.

16: Rockwood N, Abdullahi LH, Wilkinson RJ, Meintjes G. Risk Factors for

Acquired Rifamycin and Isoniazid Resistance: A Systematic Review and Meta-

Analysis. PLoS One. 2015 Sep 25;10(9):e0139017. doi:

10.1371/journal.pone.0139017. Erratum in: PLoS One. 2015;10(11):e0142276. PMID:

26406228; PMCID: PMC4583446.

17: Teo E, House H, Lockhart K, Purchuri SN, Pushparajah J, Cripps AW, van Driel

ML. Haemophilus influenzae oral vaccination for preventing acute exacerbations

of chronic bronchitis and chronic obstructive pulmonary disease. Cochrane

Database Syst Rev. 2014 Sep 9;(9):CD010010. doi: 10.1002/14651858.CD010010.pub2.

Update in: Cochrane Database Syst Rev. 2017 Jun 19;6:CD010010. PMID: 25201571.

18: Fisher-Hoch SP, Mathews CE, McCormick JB. Obesity, diabetes and pneumonia:

the menacing interface of non-communicable and infectious diseases. Trop Med Int

Health. 2013 Dec;18(12):1510-9. doi: 10.1111/tmi.12206. Epub 2013 Oct 15. PMID:

24237786.

19: Graham S, Guy RJ, Cowie B, Wand HC, Donovan B, Akre SP, Ward JS. Chronic

hepatitis B prevalence among Aboriginal and Torres Strait Islander Australians

since universal vaccination: a systematic review and meta-analysis. BMC Infect

Dis. 2013 Aug 31;13:403. doi: 10.1186/1471-2334-13-403. PMID: 24004727; PMCID:

PMC3846608.

20: van Assen S, Elkayam O, Agmon-Levin N, Cervera R, Doran MF, Dougados M,

Emery P, Geborek P, Ioannidis JP, Jayne DR, Kallenberg CG, Müller-Ladner U,

Shoenfeld Y, Stojanovich L, Valesini G, Wulffraat NM, Bijl M. Vaccination in

adult patients with auto-immune inflammatory rheumatic diseases: a systematic

literature review for the European League Against Rheumatism evidence-based

recommendations for vaccination in adult patients with auto-immune inflammatory

rheumatic diseases. Autoimmun Rev. 2011 Apr;10(6):341-52. doi:

10.1016/j.autrev.2010.12.003. Epub 2010 Dec 20. PMID: 21182987.

21: van Assen S, Agmon-Levin N, Elkayam O, Cervera R, Doran MF, Dougados M,

Emery P, Geborek P, Ioannidis JP, Jayne DR, Kallenberg CG, Müller-Ladner U,

Shoenfeld Y, Stojanovich L, Valesini G, Wulffraat NM, Bijl M. EULAR

recommendations for vaccination in adult patients with autoimmune inflammatory

rheumatic diseases. Ann Rheum Dis. 2011 Mar;70(3):414-22. doi:

10.1136/ard.2010.137216. Epub 2010 Dec 3. PMID: 21131643.

22: Mosca M, Tani C, Aringer M, Bombardieri S, Boumpas D, Brey R, Cervera R,

Doria A, Jayne D, Khamashta MA, Kuhn A, Gordon C, Petri M, Rekvig OP, Schneider

M, Sherer Y, Shoenfeld Y, Smolen JS, Talarico R, Tincani A, van Vollenhoven RF,

Ward MM, Werth VP, Carmona L. European League Against Rheumatism recommendations

for monitoring patients with systemic lupus erythematosus in clinical practice

and in observational studies. Ann Rheum Dis. 2010 Jul;69(7):1269-74. doi:

10.1136/ard.2009.117200. Epub 2009 Nov 5. PMID: 19892750; PMCID: PMC2952401.

23: Azzopardi P, Bennett CM, Graham SM, Duke T. Bacille Calmette-Guérin vaccine-

related disease in HIV-infected children: a systematic review. Int J Tuberc Lung

Dis. 2009 Nov;13(11):1331-44. PMID: 19861003.

24: Blomberg B. Antibiotikaresistens i utviklingsland [Antimicrobial resistance

in developing countries]. Tidsskr Nor Laegeforen. 2008 Nov 6;128(21):2462-6.

Norwegian. PMID: 19096470.

25: Destefano F, Pfeifer D, Nohynek H. Safety profile of pneumococcal conjugate

vaccines: systematic review of pre- and post-licensure data. Bull World Health

Organ. 2008 May;86(5):373-80. doi: 10.2471/blt.07.048025. PMID: 18545740; PMCID:

PMC2647448.

English is OK but can be improved; 

Author Response

Authors wish to thank reviewer 2 for the precious comments.

Authors' reply: reviewer 2 provided a large amount of references to be considered for being included in manuscript; since a significant number of them focused on general aspects of population vaccination (rather than HPV vaccination), the authors selected a number of references, in their opinion relevant to the manuscript's scope, and included them in the text and reference list.

Reviewer 3 Report

1. This is a small descriptive paper. Further mining data to enhance the scientificity of the paper.

2. List statistical analysis in a separate chapter. Provide a detailed introduction to the statistical analysis methods used in this article.

3. Draw a table according to the requirements of the journal. It is recommended to use a three line table.

4. Statistical analysis is needed to determine whether there is a significant difference in the distribution of opportunistic vaccination among different populations. The statistical analysis results need to be labeled in the corresponding positions.

no

Author Response

The Authors wish to thank reviewer 3 for the precious comments and reply as follows.

  1. Authors are aware that their article may be considered a "small descriptive" paper and they never thought or had the ambition to have submitted an Award Winning Paper. The issue of HPV opportunistic vaccination can be approached in different ways: virological, immunological, oncological, social, economical, an so on. In the authors' view, the option of choosing the socio-demographic aspects seemed to be interesting and informative for readers and clinicians. Obviously, the value of the manuscript can be improved and for that, reviewer 3 comments have been extensively taken into consideration.
  2. Statistical analysis has been detailed in the text and performed where possible,  due to data availability and correctness of comparison.
  3. A new table was considered redundant as data are clearly reported in the original tables 1 and 2
  4. Statistical analysis is now clearly identifiable and labeled in the respective positions.

Round 2

Reviewer 1 Report

The authors have adequately addressed the issues raised in the first round of review, and the manuscript has been revised accordingly. I have no further comments on the manuscript and recommend that it be accepted for publication.

Author Response

Authors wish to thank reviewer 1 for approving the revised manuscript

Reviewer 2 Report

It is fine now

Author Response

Authors wish to thank reviewer 2 for approving the revised manuscript

Reviewer 3 Report

1.According to the reviewers' suggestions, the author made some modifications to the paper.

2. The author believes that the formats of Table 1 and table 2 need not be modified. The editor is requested to determine whether it meets the requirements of this journal.

3.It is necessary to statistically analyze whether there are significant differences between the data of different ages.

Minor editing of English language required.

Author Response

Further statistical analysis added according to reviewer's request

Further editing of English language performed